# Morphological and Molecular Characterization of Five Species Including Three New Species of Golden Gorgonians (Cnidaria: Octocorallia) from Seamounts in the Western Pacific

**DOI:** 10.3390/biology10070588

**Published:** 2021-06-26

**Authors:** Yu Xu, Zifeng Zhan, Kuidong Xu

**Affiliations:** 1Laboratory of Marine Organism Taxonomy and Phylogeny, Shandong Province Key Laboratory of Experimental Marine Biology, Center for Ocean Mega-Science, Institute of Oceanology, Chinese Academy of Sciences, Qingdao 266071, China; xuyu16@mails.ucas.ac.cn (Y.X.); zzhan@qdio.ac.cn (Z.Z.); 2Southern Marine Science and Engineering Guangdong Laboratory (Zhuhai), Zhuhai 519082, China; 3University of Chinese Academy of Sciences, Beijing 100049, China

**Keywords:** Anthozoa, Chrysogorgiidae, *Iridogorgia flexilis*, *Iridogorgia verrucosa*, *Iridogorgia densispiralis*, taxonomy, morphology, phylogeny

## Abstract

**Simple Summary:**

Deep-water octocorals are main components of vulnerable marine ecosystems (VMEs) and play an important role in conservation and research. *Iridogorgia* Verill, 1883 is a distinct octocoral group characterized by a remarkably spiral structure and large size in deep sea, where the diversity of *Iridogorgia* is poorly known in the Western Pacific. Based on the collection of *Iridogorgia* specimens from seamounts in the tropical Western Pacific, we described five species including three new species using an integrated morphological-molecular approach. We assessed the potential of the mitochondrial genes MutS and COI, and the nuclear 28S rDNA for species delimitation and phylogenetic reconstruction of *Iridogorgia*. The results suggest that the mitochondrial markers are not able to resolve the species boundaries and deeply divergent relationships adequately, while 28S rDNA showed potential application in DNA barcoding and phylogenetic reconstruction for this genus. This study will help to understand the *Iridogorgia* biodiversity in the Western Pacific and shed more light on the screening of barcodes for octocorals.

**Abstract:**

Members of genus *Iridogorgia* Verrill, 1883 are the typical deep-sea megabenthos with only seven species reported. Based on an integrated morphological-molecular approach, eight sampled specimens of *Iridogorgia* from seamounts in the tropical Western Pacific are identified as three new species, and two known species *I. magnispiralis* Watling, 2007 and *I. densispicula* Xu, Zhan, Li and Xu, 2020. *Iridogorgia flexilis* sp. nov. is unique in having a very broad polyp body base with stout and thick scales. *Iridogorgia densispiralis* sp. nov. can be distinguished by rods present in both polyps and coenenchyme, and *I. verrucosa* sp. nov. is characterized by having numerous verrucae in coenenchyme and irregular spindles and scales in the polyp body wall. Phylogenetic analysis based on the nuclear 28S rDNA indicated that *I. densispiralis* sp. nov. showed close relationships with *I. splendens* Watling, 2007 and *I. verrucosa* sp. nov., and *I. flexilis* sp. nov. formed a sister clade with *I. magnispiralis*. In addition, due to *Rhodaniridogorgia fragilis* Watling, 2007 nested into the *Iridogorgia* clade in mtMutS-COI trees and shared highly similar morphology to the latter, we propose to eliminate the genus *Rhodaniridogorgia* by establishing a new combination *Iridogorgia fragilis* (Watling, 2007) comb. nov. and resurrecting *I. superba* Nutting, 1908.

## 1. Introduction

Deep-water octocorals are main components of vulnerable marine ecosystems (VMEs) and represent one group of VME-indicator species, playing an important role in conservation and research [1,2]. The octocorals usually have slow growth rates and long life, which suggest that full recovery from damage can take from decades to centuries, depending on the species [3]. The genus *Iridogorgia* is an easily recognized octocoral group which is characterized by a remarkably spiral structure and large size at water depths of 558–2311 m [4,5,6,7]. The species *Iridogorgia magnispiralis* Watling, 2007 is a representative, which is known as the world’s largest gorgonian reaching about 5.7 m tall [8]. 

Currently, only seven valid species are known in the genus *Iridogorgia* [7,9]. Among them, four species are found in the North Atlantic, especially on the seamounts of the Northwest Atlantic, including *I. pourtalesii* Verrill, 1883, *I. magnispiralis* Watling, 2007, *I. splendens* Watling, 2007 and *I. fontinalis* Watling, 2007 [4,10,11]. Two species are recorded off the coast of Hawaii: *I. bella* Nutting, 1908 and *I. magnispiralis* [4,8,11,12,13]. The diversity of *Iridogorgia* in the Western Pacific is still poorly known, *I. magnispiralis* was found in the Phoenix area and Northeast Australia [6,14]*,* and *I. densispicula* Xu, Zhan, Li and Xu, 2020 and *I. squarrosa* Xu, Zhan, Li and Xu, 2020 are found in the seamounts near the Mariana Trench [7].

While studying the seamount benthic diversity in the tropical Western Pacific Ocean, we collected eight specimens of *Iridogorgia* from the seamounts located on the Caroline Ridge in 2019. Based on the integrated morphological-molecular approach, we described three new species including *Iridogorgia flexilis* sp. nov., *I. verrucosa* sp. nov. and *I. densispiralis* sp. nov., and reported two known species *I. magnispiralis* and *I. densispicula*. Meanwhile, we assessed the potential of the genetic distance of the mitochondrial genes MutS and COI, and the nuclear 28S rDNA for species discrimination of this genus. We also compared the resolving power of the mitochondrial and nuclear markers for the phylogenetic reconstruction. The objective of this study was to help understand the *Iridogorgia* diversity in the Western Pacific and shed more light on the screening of barcodes for the octocorals.

## 2. Materials and Methods

### 2.1. Specimen Collection and Morphological Examination

Eight specimens were obtained from the seamounts located on the Caroline Ridge by ROV FaXian (Discovery) in the tropical Western Pacific in 2019. These specimens were photographed in situ before being sampled, photographed on board, and then stored in 75% ethanol after collection. A few branches were stored at −80 °C for molecular research. Their morphology and anatomy were examined by using a stereo dissecting microscope. The preparation of sclerites for imaging follows Xu et al. (2020) [7]. SEM scans were obtained and the optimum magnification was chosen for each kind of polyps and sclerites by using TM3030Plus SEM. The morphological terminology used follows Bayer et al. (1983) [15].

The type specimen of the three new species has been deposited in the Marine Biological Museum of Chinese Academy of Sciences (MBMCAS) Qingdao, China. The three new species have been registered in ZooBank.

### 2.2. DNA Extraction and Sequencing

Genomic DNA was extracted by a DNeasy Blood and Tissue Kit (Qiagen, Hilden, Germany) following the instructions. The mitochondrial regions of mutS homolog (mtMutS) and cytochrome oxidase subunit I (COI) and an approximately 800-nt fragment of the 28S nuclear ribosomal gene (28S rDNA) were selected for the phylogenetic analysis. To amplify mtMutS and COI, the primer pairs AnthoCorMSH (5′-AGGAGAATTATTCTAAGTATGG-3′) [16] and Mut-3458R (5′-TSGAGCAAAAGCCACTCC-3′) [17], COI8414-F (5′-CCAGGTAGTATGTTAGGRGA-3′; McFadden, unpublished) and HCO2198 (5′- TAAACTTCAGGGTGACCAAAAAATCA-3′) [18] were used, respectively. PCR reactions were conducted with the following conditions: 2 min at 98 °C followed by 35 cycles (98 °C for 15 s, 45–50 °C for 15 s, and 72 °C for 60 s) and a final extension at 72 °C for 2 min. We sequenced 28S rDNA using primers 28S-Far (5′-CACGAGACCGATAGCGAACAAGTA-3′) and 28S-Rar (5′-TCATTTCGACCCTAAGACCTC-3′) [19], and the same PCR protocol used for mtMutS. PCR reactions were performed using I-5^TM^ 2 × High-Fidelity Master Mix DNA polymerase, and sequencing was performed by TsingKe Biological Technology (TsingKe Biotech, Beijing, China).

### 2.3. Genetic Distance and Phylogenetic Analyses

All the *Iridogorgia* sequences and the related chrysogorgiid genera and out-group species *Plexaura kuna* were downloaded from GenBank (Table 1). The sequences were aligned using MAFFT v.7 [20] with the G-INS-i and Q-INS-i algorithms for the mitochondrial and 28S rDNA regions, respectively. The nucleotide alignment was trimmed to equal length using BioEdit v7.0.5 [21]. The COI alignment showed that *Iridogorgia magnispiralis* DQ860111 and *I. splendens* DQ860112 shared few overlap positions (<100 bp) with the other *Iridogorgia* sequences, and therefore these two sequences were excluded from the analysis. Genetic distances of single loci and the concatenated region mtMutS-28S between species/populations were calculated with MEGA 6.0 using Kimura 2-parameter model [22].

Phylogenetic analyses were performed on 28S rDNA and the concatenated mtMutS–COI. For the phylogenetic construction, when conspecific sequences showed no genetic variation, one was chosen randomly for analysis. Model selection and Maximum likelihood (ML) analyses were carried out using PhyML (ver. 3.0, see http://www.atgc-montpellier.fr/phyml/(accessed on 28 April 2021); [23]) on the online ATGC bioinformatic platform. The best-fitted model GTR + G was selected with the Akaike information criterion for both the mitochondrial and 28S rDNA alignments. Node support for the ML trees came from a majority-rule consensus tree of 1000 bootstrap replicates. Following Hillis and Bull (1993) [24], the ML bootstraps <70%, 70–94% and ≥95% were considered as low, moderate and high, respectively. Bayesian inference (BI) analysis was carried out using MrBayes v3.2.7a [25]. Posterior probability was estimated based on 10,000,000 Monte Carlo Markov Chain (MCMC) generations (×4 chains) sampling every 1000 generations (burn-in = 25%). Convergence of the MCMC was assessed using Tracer 1.4.1 [26]. Following Alfaro et al. (2003) [27], the Bayesian posterior probabilities <0.95 and ≥0.95 were considered as low and high, respectively.
biology-10-00588-t001_Table 1Table 1The sequences used in this study.SpeciesVoucher NumberLocatiomReferencesGenBank Accession NumbersmtMutSCOI28S rDNA*Iridogorgia densispiralis* sp. nov.MBM28645410.37° N, 140.30° Epresent study**MW841033****MW841036****MW841043***Iridogorgia flexilis* sp. nov.MBM28645310.40° N, 140.90° Epresent study**MW841031****MW841035****MW841041***Iridogorgia verrucosa* sp. nov.MBM28645510.37° N, 140.04° Epresent study**MW841034****MW840138****MW841044***Iridogorgia densispicula*MBM28653810.35° N, 140.07° E[7]MK431864**MW841037****MW841040***Iridogorgia magnispiralis*MBM28645010.50° N, 140.11° Epresent study**MW841032****MW841039****MW841042***Iridogorgia squarrosa*MBM28653911.16° N, 139.25° E[7]MK431865––*Iridogorgia fontinalis*YPM 3858434.81° N, 50.50° W[6]EU293802GQ868321–*Iridogorgia magnispiralis*YPM: IZ: 38580–France, unpublishedDQ860108––*Iridogorgia magnispiralis*MNHN-Oct.0000-057638.47° N, 27.9° W[6]GQ353316––*Iridogorgia magnispiralis*–38.78° N, 63.96° W[6]EU268055––*Iridogorgia magnispiralis*–unknownFrance & Pante, unpublished–FJ268639–*Iridogorgia magnispiralis*YPM 3858038.78° N, 63.96° W[6]JN227997––*Iridogorgia magnispiralis*–35.19° N, 47.68° W[6]GQ223116GQ868318–*Iridogorgia magnispiralis*YPM 3858138.26° N, 60.55° W[6]GQ180141––*Iridogorgia magnispiralis*USNM 109226534.58° N, 56.84° W[6]GQ180142––*Iridogorgia magnispiralis*YPM 3858238.86° N, 63.91° W[6]GQ180140––*Iridogorgia magnispiralis*–Gulf of Mexico[28,29]KC788263KC788237KX890214*Iridogorgia* sp.–21.32° N, 157.02° W[6]GQ868342GQ868323–*Iridogorgia* sp.YPM 2886633.79° N, 62.59° W[6]DQ297422––*Iridogorgia* sp. type C–23.05° N, 163.16° W[6]JN227919––*Iridogorgia* sp. type AMNHN-IC.2009-00019.15° S, 158.27° E[6]GQ180145––*Iridogorgia splendens*YPM 35397, 3858638.85° N, 63.76° W[6]DQ860109––*Iridogorgia splendens*USNM 109226738.79° N, 64.13° W[6]JN227996GQ868313–*Iridogorgia splendens*YPM 3858537.46° N, 59.95° W[6]JN228005GQ868330–*Iridogorgia splendens*–Gulf of Mexico[28]KC788271KC788229–*Iridogorgia splendens*–Gulf of Mexico[29]––KX890215*Iridogorgia splendens*USNM 1092267Kelvin Seamount, NW Atlantic[30]GQ180143––*Iridogorgia splendens*YPM: IZ: 3858537.46° N, 59.95° W[30]GQ180144––*Rhodaniridogorgia fragilis*YPM 3858834.46° N, 56.73° W[6]JN228000JN227954–*Chrysogorgia averta*–Gulf of Mexico[28]KC788265KC788235KC788258*Chrysogorgia* sp.–Gulf of Mexico[28]KC788268KC788223KC788240*Chrysogorgia* sp.–Gulf of Mexico[29]––KX890212*Pseudochrysogorgia bellona*MNHN-IC.2008-007 Paratype21.12° S, 158.5° E[6]GQ868332GQ868310–*Pseudochrysogorgia* sp.XMUB769715.52° N, 110.96° E[31]––MW336977*Metallogorgia melanotrichos*–34.53° N, 47.79° W[6]GQ180158FJ268633–*Metallogorgia macrospina*NIWA1564237.21° S, 177.24° E[6]JN228001JN227952–*Radicipes stonei*USNM: IZ:1418007Alaska[32]MG986912MG986961MG980134*Stephanogorgia faulkneri*NTM C0149277.19° N, 134.32° E[6]GQ342485GQ342406JX203718Chrysogorgiidae sp.–41.37° S, 42.85° E[33]KP324387KP678004KP324611*Plexaura kuna*RMNH Coel.40836South Africa[34]JX203807JX203866JX203748New sequences are in bold.

## 3. Results

### 3.1. Systematics

Class Anthozoa Ehrenberg, 1834 [35].

Subclass Octocorallia Haeckel, 1866 [36].

Order Alcyonacea Lamouroux, 1812 [37].

Suborder Calcaxonia Grasshoff, 1999 [38].

Family Chrysogorgiidae Verrill, 1883 [10].

Genus *Iridogorgia* Verrill, 1883.

Diagnosis (based on Watling, 2007 [4]). Chrysogorgiids with main axis monopodial and spiraling upward, and undivided branches emanating from one side of axis. Polyps uniserially arranged, when sexually mature with base expanded along upper side of branch. Sclerites rods, spindles, or scales, extending in tracts onto tentacles. Branch coenenchyme with sclerites oriented along branch or absent between polyps.

Type species. *Iridogorgia pourtalesii* Verrill, 1883, by monotypy.

Distribution. Pacific: tropical Western Pacific, Hawaii, Southwest Pacific; Atlantic: North and Central East Atlantic; Indian Ocean: Great Australian Bay, depths 558–2311 m [4,5,6,7,8,10,11,12,13,14,39,40].

*Iridogorgia flexilis* sp. nov.

Figure 1A–G, and Figure 2; Table 2.

urn:lsid:zoobank.org:act:6BD2C9FC-9CD8-4604-971D-B2EC0BB108B8.

Material examined. Holotype: MBM286453, station FX-Dive 222 (10°4.7′ N, 140°9.28′ E), a seamount (temporarily named as M5) on the Caroline Ridge, 2016 m, 10 June 2019.

Diagnosis. Colony having a long unbranched part with branches producing on the top. Axis loosely coiled on the top with helical turn 12–15 cm high. Polyps having a broad body base with its width usually longer than the height. Rods in the back of tentacle rachis longitudinally arranged, slender with more or less warts. Spindles and scales in polyp body wall transversely or obliquely arranged, stout and thick, occasionally crossed with irregular edges. Spindles in coenenchyme slender and thick, occasionally branched. Verrucae rare in branches.

Description. Holotype is incomplete colony without holdfast, about 85 cm long (Figure 1B). The direction of growth clockwise. Axis about 4 mm in diameter at base with iridescent metallic luster. The branching part about 34 cm long including three loosely helical turns with each helical turn 12–15 cm long and 3–4 cm in diameter. Branches arranged along one side, about 3 mm apart, up to 28 cm long with 39 polyps counted. Polyps have a broad body base with its width usually longer than the height, forming a contraction at the base of the tentacular part (Figure 1C,D). Polyps 4–6 mm apart, 1–3 mm tall, and 2–4 mm wide at base (Figure 1a). Tentacular part about 1 mm long, 1.0–1.5 mm wide, composed of numerous rods usually forming eight obvious parallel columns terminating at its base. Polyps white after fixation in alcohol. Verrucae rare in branches.

Rods in the back of tentacle rachis longitudinally arranged, slender with two round ends, surface usually with numerous small conical or ridge-like warts, occasionally with sparse warts or shallow cracks, measuring 261–532 × 19–58 μm (Figure 1F and Figure 2A). Spindles and scales in polyp body wall transversely or obliquely arranged, stout and thick, occasionally crossed with irregular edges or various shapes, measuring 172–679 × 27–219 μm (Figure 1G and Figure 2B). Their surface usually with sparse and fine warts, occasionally nearly smooth or with large protuberances and some shallow cracks. Spindles in coenenchyme slender and thick, usually with two rounded ends, nearly smooth or with sparse and fine warts, and occasionally branched or with narrow or sharp ends, measuring 322–892 × 25–72 μm (Figure 1E and Figure 2C).

Type locality. A seamount (temporarily named as M5) located on the Caroline Ridge in the Western Pacific with water depth of 2016 m.

Etymology. The Latin adjective *flexilis* (flexile) refers to the flexile axis of this species.

Distribution and Habitat. Found only from a seamount located on the Caroline Ridge. Grown in a rocky bottom with the water temperature about 2.1 °C and salinity 36.5 (Figure 1A).

Remarks. *Iridogorgia flexilis* sp. nov. is characterized by its highly helical turns, broad polyp body base with stout and thick spindles and scales. It is similar to *I. magnispiralis* Watling, 2007 and *I. densispicula* Xu et al., 2020 in the helical turn, but differs distinctly from *I. magnispiralis* by the presence of scales in polyp body wall (vs. absent), from *I. densispicula* by the thick and relatively smooth sclerites in the polyp body wall (vs. very thin and coarse often with rugged and ridged surface) [4,7].

*Iridogorgia densispiralis* sp. nov.

Figure 1b, Figure 3 and Figure 4; Table 2.

urn:lsid:zoobank.org:act:026E2F14-7A66-4558-9D65-B7E3CC0EC8D0.

Material examined. Holotype: MBM286454, station FX-Dive 225 (10°36.75′ N, 140°3.85′ E), a seamount (temporarily named as M8) located on the Caroline Ridge, 1574 m, 13 June 2019.

Diagnosis. Colony relatively short. Axis have many close helical turns each 2.5–3.0 cm high. Polyps small with an expanded and conical body base. Rods in tentacle rachis longitudinally arranged, usually with many small warts. Spindles, rods and a few elongated scales in polyp body wall transversely or obliquely arranged, usually thick with sparse and fine warts. Rods and spindles in coenenchyme thick and nearly smooth or with sparse and fine warts. Branches with a few verrucae.

Description. Holotype orange in situ, about 48 cm long with the holdfast not recovered (Figure 3A,B). The direction of growth clockwise. Axis about 3 mm in diameter at base with branches producing nearly from the bottom to top and light iridescent metallic luster. Axis has 13 helical turns each 2.5–3.0 cm high and 0.5 cm wide. Branches usually grow upward, arranged along one side, 2–3 mm apart. Branches almost incomplete after collection and up to 11 cm long with 14 polyps counted. Polyps small, 3–9 mm apart, 1–2 mm tall and 1–3 mm wide at base (Figure 1b and Figure 3C–E). Tentacular part about 1.0–1.5 mm long and wide, situated at an acute angle with the branch. Polyps white after fixation in alcohol. A few large verrucae present at polyp base and branches.

Rods in the back of tentacle rachis longitudinally arranged, with many small warts and two rounded ends, sometimes with deep cracks or a few warts, measuring 95–442 × 11–52 μm (Figure 3F and Figure 4A). Spindles, rods and a few elongated scales in polyp body wall transversely or obliquely arranged, often thick with sparse and fine warts, occasionally branched or crossed with shallow cracks on surface, measuring 84–347 × 18–60 µm (Figure 4B). Rods and spindles in coenenchyme, thick with two sharp or round ends, nearly smooth or with sparse and fine warts, occasionally crossed, measuring 90–523 × 19–65 µm (Figure 3G and Figure 4C). Sclerites sometimes sparse in the coenenchyme between polyps.

Type locality. A seamount (tentatively named as M8) located on the Caroline Ridge in the Western Pacific with water depth of 1574 m.

Etymology. Composite of the Latin adjectives *densus* (dense) and *spiralis* (spiral), referring to the dense spirals of this species.

Distribution and Habitat. Found only from a seamount located on the Caroline Ridge. Grown in a rocky bottom with branches in upper part of the colony incomplete (Figure 3A).

Remarks. *Iridogorgia densispiralis* sp. nov. is characterized by the relatively short colony, short and narrow helical turn, and rods present in both polyps and coenenchyme. It is most similar to *I. splendens* Watling, 2007 in the relatively short colony and helical turn in adults, but differs by the presence of thick rods in coenenchyme (vs. absent) and abundant sclerites in tentacles (vs. sparse) [4].

*Iridogorgia verrucosa* sp. nov.

Figure 1c, Figure 5 and Figure 6; Table 2.

urn:lsid:zoobank.org:act:E388B7B2-936E-485C-AB38-E7F95DD4F0A4.

Material examined. Holotype: MBM286455, station FX-Dive 225 (10°36.77′ N, 140°04.05′ E), a seamount (temporarily named as M8) located on the Caroline Ridge, 1397 m, 13 June 2019.

Diagnosis. Colony relatively short with branches producing nearly from the bottom to top. Axis having many close helical turns each 3–4 cm in high. Polyps small with long tentacles. Rods in tentacle rachis longitudinally arranged and usually with many small warts. Spindles and elongated scales in polyp body wall transversely or obliquely placed, thick with irregular edges. Spindles in coenenchyme slender and thick with usually rounded ends. Branches with many cylindroid verrucae.

Description. Holotype orange, about 51 cm long with the holdfast not recovered (Figure 5B). Growth direction clockwise. Axis about 4 mm in diameter at base with branches producing nearly from bottom to top, composed of 14 helical turns each about 3–4 cm high and 1 cm wide. Branches usually growing upward, arranged along one side, 1–3 mm apart. Branches almost incomplete after collection, up to 10 cm long with 14 polyps counted. Polyps small, towards to the branch end and situated with an acute angle, 4–6 mm apart, average 2 mm tall, and 1–2 mm wide at base (Figure 1c and Figure 5C–E). Tentacular part up to 2 mm long and 1 mm wide. Polyps white after fixation in alcohol. Numerous and dense small cylindroid verrucae present in polyp base and branches (Figure 5C,G).

Rods in the back of tentacle rachis longitudinally arranged, usually with many small conical warts and round ends, sometimes with deep cracks on surface, occasionally lobed with irregular edges, measuring 160–459 × 17–73 μm (Figure 5F and Figure 6A). Spindles and elongated scales in polyp body wall transversely or obliquely placed, usually thick with sparse and fine warts and irregular edges, measuring 116–450 × 21–65 µm (Figure 6B). Spindles in coenenchyme very similar to those in polyp body wall, almost slender, measuring 169–438 × 19–62 µm (Figure 6C).

Type locality. A seamount (temporarily named as M8) located on the Caroline Ridge in the Western Pacific with water depth of 1397 m.

Etymology. The Latin adjective *verrucosus* (verrucose) refers to the numerous small verrucae of this species.

Distribution and Habitat. Found only from a seamount located on the Caroline Ridge. Grown in a rocky bottom with black coral and scleractinian coral living around (Figure 5A).

Remarks. *Iridogorgia verrucosa* sp. nov. is characterized by the relatively short colony and narrow helical turn, irregular spindles and scales in polyp body wall and numerous verrucae. It is similar to *I. splendens* and *I. densispiralis* sp. nov. in the short colony and close helix, but differs from *I. splendens* by the abundant rods in tentacles (vs. sparse), sclerites in polyp body wall with irregular edges (vs. regular), and numerous sclerites in inter-polyp coenenchyme (vs. rare to absent) [4]. *Iridogorgia verrucosa* sp. nov. can be distinguished from *I. densispiralis* sp. nov. by the absence of rods in coenenchyme (vs. present) and numerous and dense verrucae in branches (vs. sparse).

*Iridogorgia magnispiralis* Watling, 2007.

Figure 1d, Figure 7 and Figure 8; Table 2.

*Iridogorgia magnispiralis* Watling, 2007: 395–397, Figure 2 and Figure 3.

Material Examined. MBM286450 and MBM286451, station FX-Dive 216 (10°5.15′ N, 140°11.12′ E), a seamount (tentatively named as M5) located on the Caroline Ridge in the Western Pacific, 845 m, 4 June 2019.

Diagnosis (based on the present specimens and Watling, 2007 [4]). Colony with branches producing on upper part, and helical turn each 12–20 cm high. Polyps erect or slightly inclined along branches. Rods in tentacle rachis longitudinally arranged, regular with many small warts. Spindles in polyp body wall transversely or obliquely arranged, slender and nearly smooth. Spindles in coenenchyme transversely arranged, slender with two sharp ends and nearly smooth surface. Scales absent. Branches with many verrucae. Inter-polyp coenenchyme with numerous sclerites.

Description. Two specimens have similar external morphology and size, and the specimen MBM286451 was described in detail here. Colony grown in a rocky bottom with a small holdfast in situ (Figure 7A). Specimen about 63 cm long with the holdfast not recovered (Figure 7C). The direction of growth clockwise and the same in MBM286450. Axis about 2.5 mm in diameter at base with branches producing on the upper part, having two helical turns each about 12 cm high and 1 cm in diameter. Branches arranged along one side, about 2 mm apart and up to 13.5 cm long with 21 polyps counted. Polyps erect or slightly inclined along the branches, 2–4 mm apart, 1–3 mm long, and 2–3 mm wide at base (Figure 1d and Figure 7B,D–F). Tentacular part 1.0–1.5 mm long and wide. Polyps white after fixation in alcohol. Branches with many low verrucae.

Rods in the back of tentacle rachis longitudinally arranged, regular with two rounded ends, often with many small conical warts, occasionally with some shallow cracks and a few warts on surface, measuring 157–548 × 18–78 μm (Figure 8A). Spindles in polyp body wall transversely or obliquely arranged, thick, slender and usually with two sharp ends, nearly smooth or with many fine warts, occasionally rugged or with a few large protuberances, measuring 214–1054 × 28–65 µm (Figure 8B). Spindles in coenenchyme slender with two sharp ends, nearly smooth, occasionally with a few large irregular protuberances, measuring 332–1032 × 21–81 µm (Figure 8C). Scales absent in this specimen (Figure 7F and Figure 8). Inter-polyp coenenchyme with numerous sclerites.

Distribution. The western Corner Rise Seamounts to the New England Seamounts to Kelvin Seamount, and the Lost City site on the Mid-Atlantic Ridge, 1650–2400 m [4]; Hawaii, 1310–1366 m [6,8]; Phoenix Islands Protected Area, 1000–2000 m [14]; Northeast Australia, 728–777 m [6]; a seamount located on the Caroline Ridge, 845 m.

Remarks. The present specimens match well with the holotype of *Iridogorgia magnispiralis* Watling, 2007. The smaller number of the helical turns and the shorter branches suggest the two specimens are in a young stage.

*Iridogorgia densispicula* Xu, Zhan, Li and Xu, 2020.

Figure 1e–g, Figure 9, Figure 10A–E, Figure 11, Figure 12 and Figure 13; Table 2.

*Iridogorgia densispicula* Xu, Zhan, Li & Xu, 2020: 251–254, Figure 2 and Figure 3.

Material Examined. MBM286446, station FX-Dive 226 (10°38.8′ N, 140°4.15′ E), a seamount (tentatively named as M8) located on the Caroline Ridge, 1741 m, 14 June 2019. MBM286447, station FX-Dive 227 (10°37.88′ N, 140°5.62′ E), a seamount (tentatively named as M8) located on the Caroline Ridge, 1678 m, 15 June 2019. MBM286448, station FX-Dive 211 (10°2.93′ N, 140°10.48′ E), a seamount (tentatively named as M5) located on the Caroline Ridge, 1482 m, 29 May 2019.

Improved diagnosis (based on the present specimens and Xu et al. 2020 [7]). Colony slender with branches producing on the upper part. Axis having helical turns each 12–19 cm high. Polyps usually inclined towards distal end of branch. Rods in tentacle rachis longitudinally arranged, regular with many small conical or ridge-like warts on surface and two rounded ends. Spindles and scales in polyp body wall transversely or obliquely arranged, elongated with irregular edges and often rugged and ridged surface. Spindles in coenenchyme slender and usually with two sharp ends. Branch surface nearly smooth, rarely with verrucae.

Description. The three specimens are similar in external morphology, and only the specimen MBM286446 was descripted in detail here. Colony grown in a rocky bottom with a small white holdfast in situ (Figure 9A). Specimen about 73 cm long with the holdfast not recovered (Figure 9B). The direction of growth counterclockwise (clockwise in MBM286447, counterclockwise in the bottom branching part and clockwise in the upper in MBM286448). Axis about 2.5 mm in diameter at base with dark iridescent metallic luster and having six helical turns, each turn 16–18 cm long and 3–4 cm in diameter. The branching part about 19 cm long with branches growing upward and not arc-shaped. Branches arranged along one side, 3–4 mm apart, and up to 20 cm long with 29 polyps counted. Polyps usually inclined towards distal end of branch, 2–4 mm apart, average 2.5 mm long, and 2–4 mm wide at base (Figure 1e and Figure 10A). Tentacular part 1–2 mm long, 1 mm wide, usually forming eight obvious parallel columns terminating at its base (Figure 10C). Polyps sometimes with an expanded body base and golden eggs present and visible under the microscope (Figure 10A). Polyps became white after fixation. Branch surface nearly smooth, rarely with a few verrucae.

Rods in the back of tentacle rachis longitudinally arranged, slender and regular with usually two round ends and many small conical or ridge-like warts, occasionally branched and with a few large protuberances, measuring 137–519 × 11–52 μm (Figure 11A). Spindles and scales in polyp body wall transversely or obliquely arranged, elongate and often with a slight midway constriction and many fine warts, usually lobed with irregular shape and rugged and ridged surface in the upper part of the body wall (Figure 10D), and regular and flat in the basal part (Figure 10E), measuring 80–540 × 14–101 μm (Figure 11B). Spindles in coenenchyme slender, usually with two sharp ends and many fine warts on surface, occasionally nearly smooth or with a few irregular protuberances, measuring 203–967 × 11–65 μm (Figure 11C). Inter-polyp coenenchyme with numerous sclerites.

Distribution. Seamounts located on the Caroline Ridge, 1204–1741 m ([7]).

Remarks. Our specimens match well with the previous description, and they all have a slender stem, high helical turn, oblique polyps, and the same sclerite forms in tentacles and coenenchyme (Figure 9, Figure 10, Figure 11, Figure 12 and Figure 13). Considering only one holotype described before, here we make a supplementary description for this species, especially the varied sclerites in the polyp body wall, including spindles and scales, which are elongated and often with irregular edges and rugged and ridged surfaces (Figure 11B, Figure 12B and Figure 13B). It is worth noting that the elongated scales are occasionally present in the tentacles of the specimen MBM286448 (Figure 13A). These special scales may be an extension from the body extending to the tentacles. Furthermore, the polyp shapes in these specimens are variable, including bud-like (holotype and MBM286448, Figure 10C), slender with elongate body (MBM286446, Figure 10A), cylindrical with long extended tentacles (MBM286447, Figure 10B). Such difference may be a process of growth stages (mature or immature) and caused by the alcohol preservation.

### 3.2. Genetic Distance and Phylogenetic Analyses

The new sequences were deposited in GenBank (Table 1). The alignments comprised 663, 537 and 797 nucleotide positions for the mtMutS, COI and 28S rDNA regions, respectively. Based on the aligned region of 28S rDNA, the interspecific distance of *Iridogorgia* ranged from 0.32% to 6.08% (average value at 3.10%), while the intraspecific distance was 0.32%, which was calculated from only two populations of *Iridogorgia magnispiralis* (Table 3). For the concatenated region mtMutS-28S, the interspecific distance range is in the range of 0.31%–3.48%, and no intraspecific data are available (Table 3). For the mitochondrial alignments, the interspecific and intraspecific distances of *Iridogorgia* are in ranges of 0–1.33% and 0–0.33% for mtMutS, and 0–0.39% and zero for COI, respectively (Table 4 and Table 5). The genetic distances between *Rhodaniridogorgia* and *Iridogorgia* are in ranges of 0.47%–1.41% for mtMutS, and 0–0.39% for COI, which partially and fully overlap with the interspecific distances of *Iridogorgia*, respectively (Table 4 and Table 5).

The ML tree is nearly identical to the BI tree in topology for both the mtMutS-COI and 28S rDNA regions, and thus a single tree with both support values was shown for each of the markers (Figure 14 and Figure 15). In the mtMutS-COI trees, the monophyly of *Iridogorgia* was not supported due to the *Rhodaniridogorgia* nested into the *Iridogorgia* clade, and clustered with the subclade of *I. magnispiralis*, *I. fontinalis* and *Iridogorgia* sp. GQ180145 with low to moderate support (Figure 14). Within *Iridogorgia*, *Iridogorgia densispicula* branched outside of the main clade including the rest of the *Iridogorgia* species. Like the subclade of *magnispiralis*/*fontinali, I. densispiralis* sp. nov., *I. verrucosa* sp. nov. and *Iridogorgia* sp. GQ868342 also formed a subclade due to no genetic variation among the them (Table 4 and Table 5 and Figure 14). Although *I. flexilis* sp. nov., *I. splendens*, *Iridogorgia* sp. GQ868342 and the subclades *magnispiralis*/*fontinalis* and *densispiralis*/*verrucosa* branched together, their deeply divergent relationships were not resolved (Figure 14). In the 28S rDNA trees, *I. densispicula* branched early, and the rest of the *Iridogorgia* species were separated to two clades with low support (ML < 70%; BI < 0.90; Figure 15). Within Clade I, *I. densispiralis* sp. nov. and *I. splendens* branched together, followed by *I. verrucosa* sp. nov. with high support. *Iridogorgia flexilis* sp. nov. and *I. magnispiralis* formed the Clade II with high support.

## 4. Discussion

The DNA barcoding analysis of mtMutS, COI and 28S rDNA is considered as one of the first steps in an integrative identification of octocorals [19,41,42]. In the present analysis, COI exhibited much less variation than the other two loci, and no genetic variability was observed among the three new species and *I. splendens* Watling, 2007 (Table 5). Consequently, COI is less informative as a species-specific marker for octocorals. For the mtMutS, there is no barcoding gap for delimitating *Iridogorgia* species due to no genetic variability observed between *I. densispiralis* sp. nov. and *I. verrucosa* sp. nov., and between *I. fontinalis* Watling, 2007 and *I. magnispiralis* Watling, 2007, indicating its limited usefulness for the species delimitation (Table 4). In contrast, the 28S rDNA showed a higher level of genetic variation (e.g., interspecific distances 0.32–6.08% with average value of 3.10% vs. 0–1.33% at mtMutS and 0–0.39% at COI). A threshold of 0.5% can delimit all available *Iridogorgia* species except for *I. splendens* KX890215 (the distance between it and *I. densispiralis* sp. nov. is 0.32%). Like the case of 28S rDNA, the concatenated region mtMutS-28S can separate the present five species with a commonly used threshold of 0.3% (Table 3). Nonetheless, only seven *Iridogorgia* sequences of 28S rDNA are available, more data including both conspecific and congeneric sequences are needed to confirm 28S rDNA and mtMutS-28S as effective barcodes for the genus *Iridogorgia*.

The genus *Rhodaniridogorgia* was established by Watling [4] to contain the species *Rhodaniridogorgia fragilis* Walting, 2007 and *R. superba* (Nutting, 1908) transferred from *Iridogorgia*. It has a similar external shape and sclerites with *Iridogorgia* and is distinct mainly by the axis shape (wavy vs. coiled; [4]). In the mtMutS-COI trees, *Rhodaniridogorgia*
*fragilis* nested into the *Iridogorgia* clade, and showed a close relationship with *I. magnispiralis* and *I. fontinalis* with low to moderate support, and the genetic distance data do not support the separation of *R. fragilis* from *Iridogorgia* (Table 4 and Table 5; Figure 14). Furthermore, the wavy shape of the axis can be seen as a specific coiled spiral, when the helical diameter is narrow. Therefore, based on the morphological and phylogenetic data, we suggest a new combination *Iridogorgia fragilis* (Watling, 2007) comb. nov., resurrect *I. superba* Nutting, 1908 and eliminate the genus *Rhodaniridogorgia.* Here, we slightly update the diagnosed *Iridogorgia* to include the *fragilis*-like species: Chrysogorgiids with main axis monopodial, wavy or coiled spiraling upward, and undivided branches emanating from one side of the axis. Polyps uniserially arranged, when sexually mature with base expanded along upper side of branches. Sclerites rods, spindles, or scales, sometimes branched or with coarse sculptures. Branches coenenchyme with sclerites oriented along branches or without sclerites between polyps.

The present molecular phylogenetic analysis supported the assignment of the new species to the genus *Iridogorgia*. However, the mitochondrial marker MutS-COI could not resolve the deeply divergent relationships among the available *Iridogorgia* species except *I. densispicula* Xu, Zhan, Li and Xu, 2020 (Figure 14). In contrast, 28S rDNA showed better resolution for the *Iridogorgia* phylogeny. In the 28S rDNA trees, *I. densispiralis* sp. nov. showed close relationships with *I. splendens* and *I. verrucosa* sp. nov. This is consistent with their morphological characters that all these species have a relatively short colony and close helical turn in adults with branches producing nearly from the bottom to top (Figure 3B, Figure 5B and Figure 15). *Iridogorgia flexilis* sp. nov. formed a sister clade with *I. magnispiralis*, which is also consistent with the morphological data that all have a high colony and loosely helical turn with branches producing on the upper part of the colony (Figure 1B, Figure 7C and Figure 15).

Now there are twelve species in the genus *Iridogorgia*, for their distributions see the Figure 16. Among these, six species were reported from the Western Pacific, including *I. flexilis* sp. nov., *I. verrucosa* sp. nov., *I. densispiralis* sp. nov., *I. magnispiralis*, *I. densispicula* and *I. squarrosa* Xu, Zhan, Li and Xu, 2020. The data indicate a high diversity of *Iridogorgia* in Western Pacific and provide a potential reference for vulnerable marine ecosystems and seamount conservation.

In order to distinguish the *Iridogorgia* species better, a dichotomous key based on morphological features is given. Due to the incomplete holotype and vague original description, *I. bella* is temporarily not considered here.

1. Colony with a wavy axis……………………………………………………………………..2

– Colony with a coiled axis ……………………………………………………….…………....3

2. Sclerites with a wide range of shapes, including spindles and scales……….…..*I. fragilis*

– Sclerites uniform in size and shape, including slender smooth rods……………*I. superba*

3. Colony with loose helix in adult, each helical turn ≥12 cm high…………...............….…4

– Colony with close helix in adult, each helical turn ≤7 cm high…………….…...…...….. 7

4. Scales present in the polyp body wall…………………………………………...….….……5

–Scales absent in the polyp body wall………………………..…..……….……*I. magnispiralis*

5. Sclerites in the polyp body wall usually without large tuberculate warts; small lumpy sclerites absent…………………………………………………………………...………….……6

– Sclerites in the polyp body wall usually sculptured by numerous large tuberculate warts; small lumpy sclerites often present…………………………..……………..*I. squarrosa*

6. Sclerites in the polyp body wall relatively slender and thin, surface coarse and/or rugged……………………………………..……...……………………………….…... *I. densispicula*

– Sclerites in the polyp body wall stout and thick, surface relatively smooth……………………………………………………………………………....….. *I. flexilis* sp. nov.

7. Sclerites rare to absent in the inter-polyps coenenchyme…………………………………8

– Sclerites sparse to numerous in the inter-polyps coenenchyme……………………..……9

8. Scales present in polyps…………….....………………………..……...…………*I. splendens*

– Scales absent in polyps……………….....………………………………...….…. *I. pourtalesii*

9. Branches producing on the top of colony (umbrella-shaped) ……........…..…*I. fontinalis*

– Branches producing from nearly bottom to top or the upper half of colony (tree-shaped) …………………………………………………………………………………………..10

10. Rods present in coenenchyme…………………………………… *I. densispiralis* sp. nov.

– Rods absent in coenenchyme………………………………….....…….. *I. verrucosa* sp. nov.

## 5. Conclusions

Based on an integrated morphological-molecular approach, eight sampled specimens of *Iridogorgia* from seamounts in the tropical Western Pacific are identified as three new species *I. flexilis* sp. nov., *I. densispiralis* sp. nov. and *I. verrucosa* sp. nov., and two known species *I. magnispiralis* Watling, 2007 and *I. densispicula* Xu, Zhan, Li and Xu, 2020. Phylogenetic analysis based on the nuclear 28S rDNA, indicated that *I. densispiralis* sp. nov. showed close relationships with *I. splendens* Watling, 2007 and *I. verrucosa* sp. nov., and *I. flexilis* sp. nov. formed a sister clade with *I. magnispiralis*. The genetic analysis demonstrates that neither the mtMutS nor COI barcodes could resolve species boundaries adequately, while 28S rDNA showed potential application in DNA barcoding and phylogenetic reconstruction for this genus. In addition, based on morphological characters and the mitochondrial gene data, we propose to eliminate the genus *Rhodaniridogorgia* by establishing a new combination *Iridogorgia fragilis* (Watling, 2007) comb. nov. and resurrecting *I. superba* Nutting, 1908.

## Figures and Tables

**Figure 1 biology-10-00588-f001:**
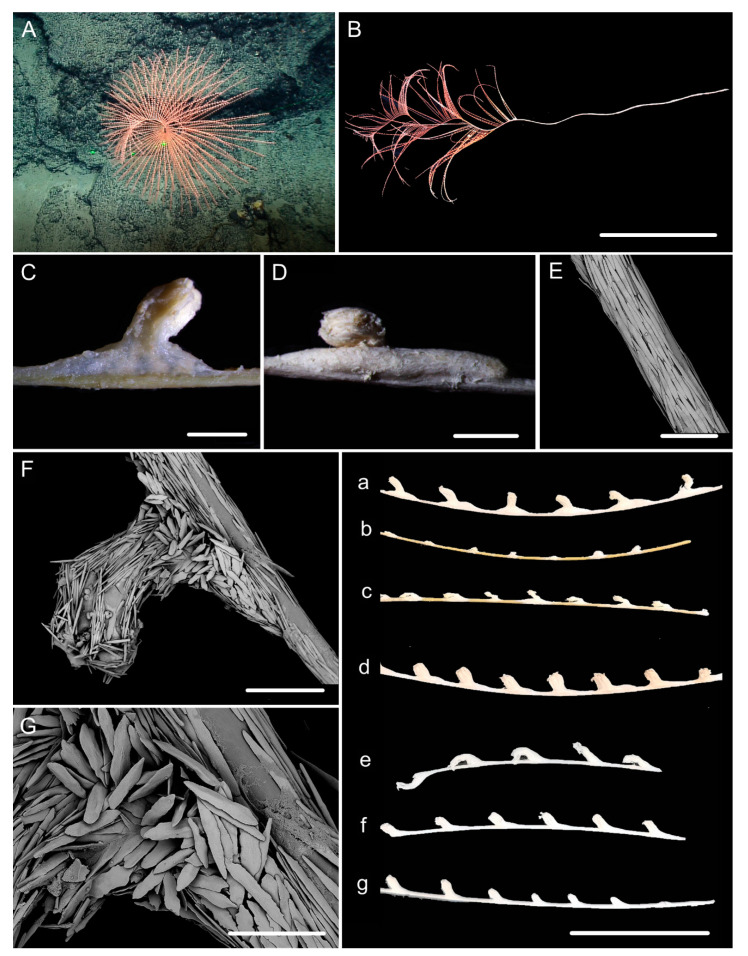
External morphology and polyps of *Iridogorgia flexilis* sp. nov. and different branch morphology of the *Iridogorgia* species in this passage (**a**–**g**). (**A**,**B)** The holotype in situ and after collection. Laser dots spaced at 33 cm used for measuring dimensions. (**C**,**D**) Single polyps under a light microscope. (**E**) A part of branch under SEM. (**F**) A single polyp under SEM. (**G**) Sclerites in basal polyp body under SEM. (**a**–**g**) A part of branch with different polyp size and arrangement, followed by *I. flexilis* sp. nov., *I. densispiralis* sp. nov., *I. verrucosa* sp. nov., *I. magnispiralis*, *I. densispicula* (MBM286446, 286447, 286448). Scales = 20 cm (**B**), 2 cm (a–g in the same scale), 2 mm (**C**,**D**), 1 mm (**F**), 500 μm (**E**,**G**).

**Figure 2 biology-10-00588-f002:**
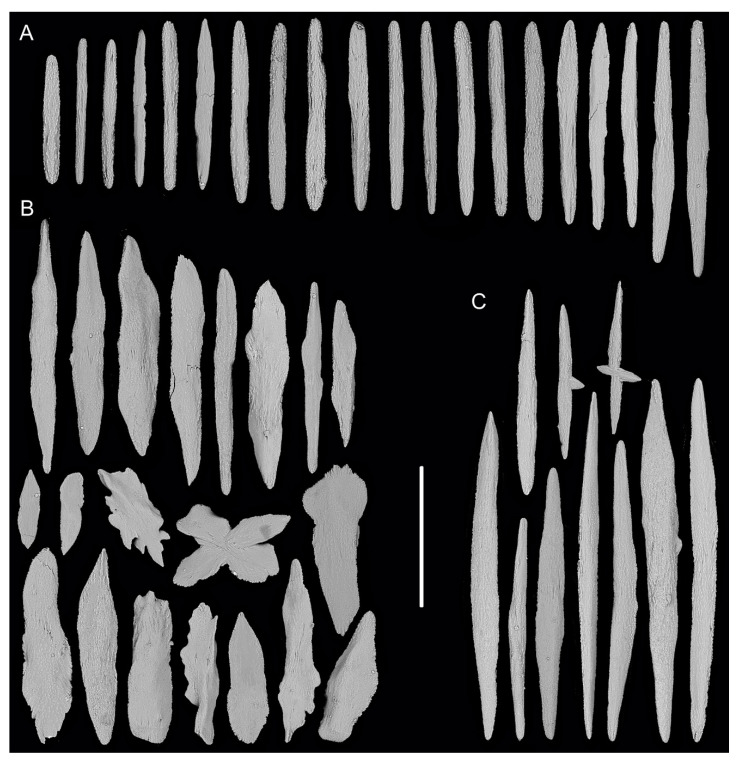
Sclerites of *Iridogorgia flexilis* sp. nov. (**A**) Sclerites in tentacle rachis. (**B**) Sclerites in polyp body wall. (**C**) Sclerites in coenenchyme. Scales = 300 μm, all in the same scale.

**Figure 3 biology-10-00588-f003:**
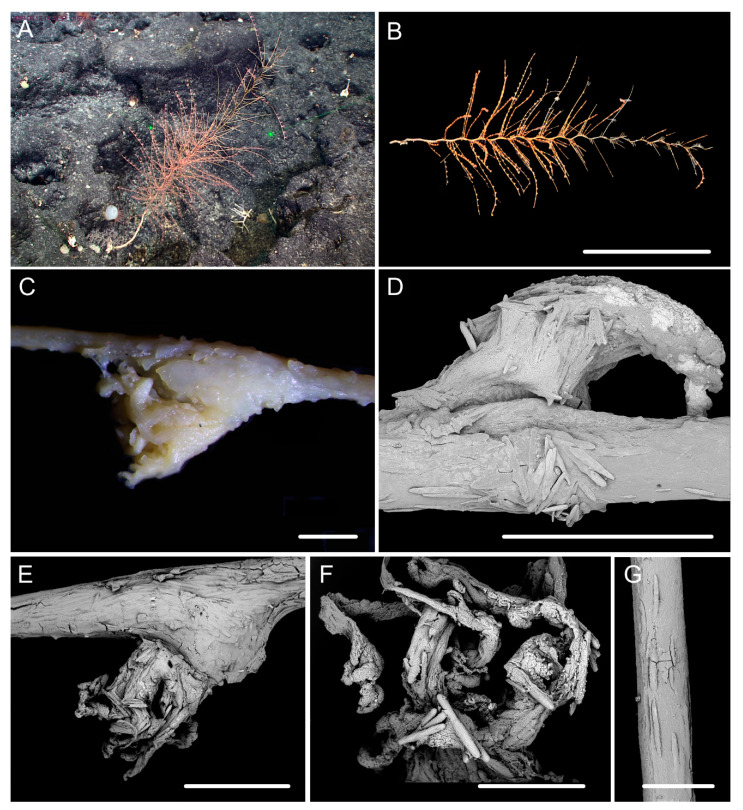
External morphology and polyps of *Iridogorgia densispiralis* sp. nov. (**A**) The holotype in situ. (**B**) The holotype after collection. (**C**) A single polyp under a light microscope. (**D,E**) Singles polyps under SEM. (**F**) Tentacular part under SEM. (**G**) A part of branch under SEM. Scales = 20 cm (**B**), 1 mm (**C**–**E**), 500 μm (**F**,**G**).

**Figure 4 biology-10-00588-f004:**
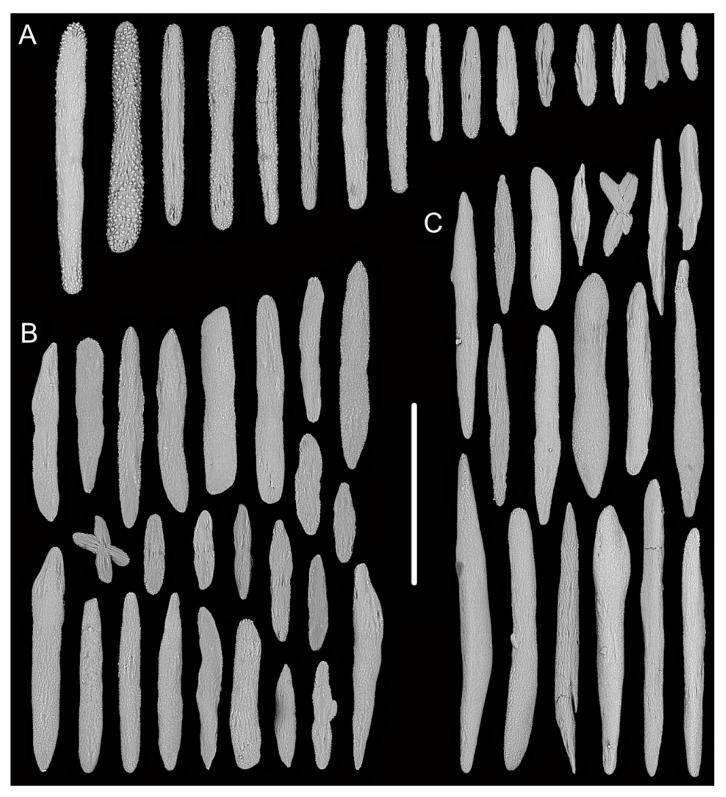
Sclerites of *Iridogorgia densispiralis* sp. nov. (**A**) Sclerites in tentacle rachis. (**B**) Sclerites in polyp body wall. (**C**) Sclerites in coenenchyme. Scales = 300 μm, all in the same scale.

**Figure 5 biology-10-00588-f005:**
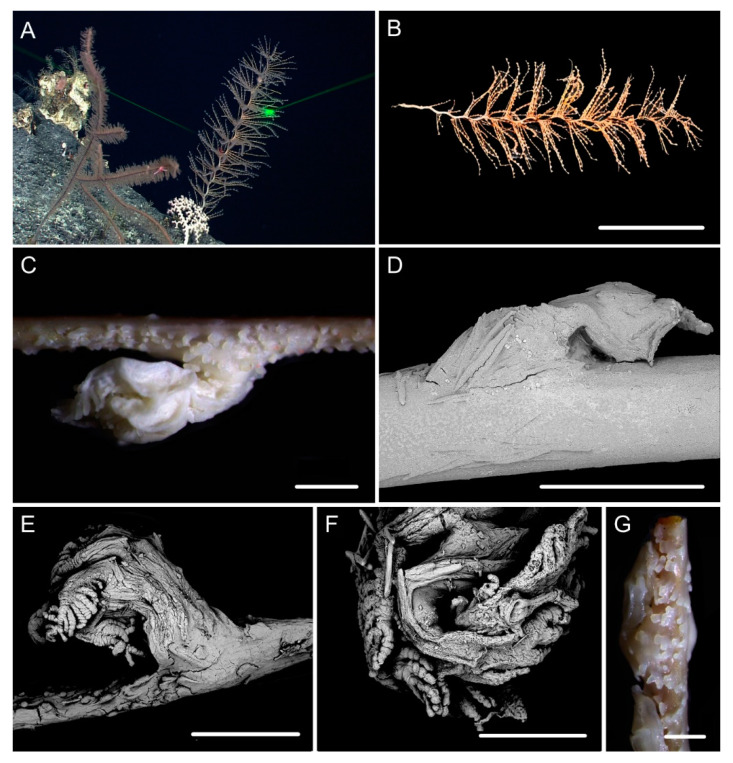
External morphology and polyps of *Iridogorgia verrucosa* sp. nov. (**A**) The holotype in situ. (**B**) The holotype after collection. (**C**) A single polyp under a light microscope. (**D**,**E**) Single polyps under SEM. (**F**) Tentacular part under SEM. (**G**) Coenenchyme with many verrucae under a light microscope. Scales = 20 cm (**B**), 1 mm (**C**–**E**,**G**), 500 μm (**F**).

**Figure 6 biology-10-00588-f006:**
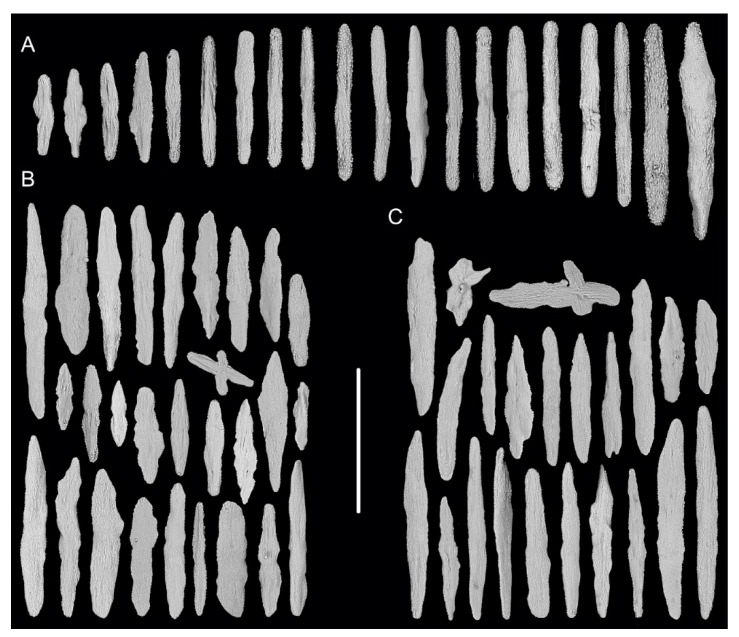
Sclerites of *Iridogorgia verrucosa* sp. nov. (**A**) Sclerites in tentacle rachis. (**B**) Sclerites in polyp body wall. (**C**) Sclerites in coenenchyme. Scales = 300 μm, all in the same scale.

**Figure 7 biology-10-00588-f007:**
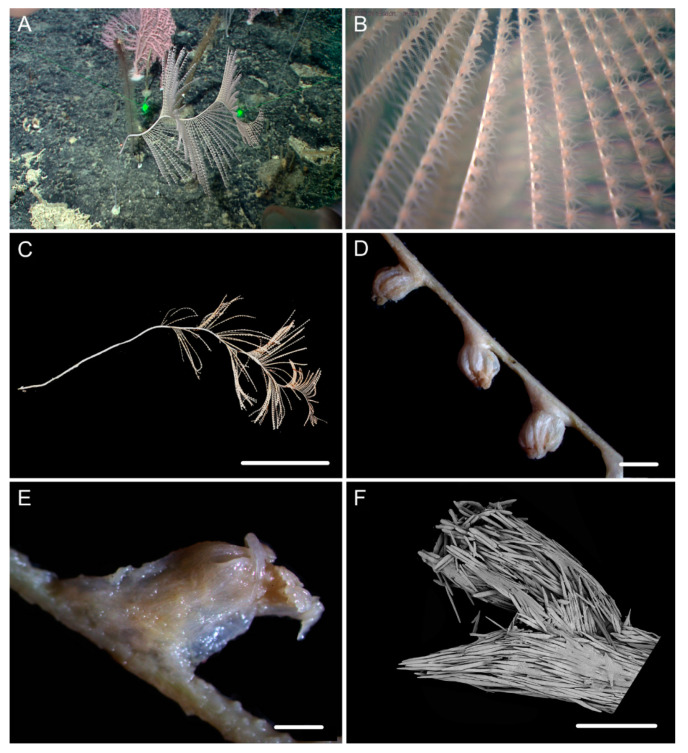
External morphology and polyps of *Iridogorgia magnispiralis* Watling, 2007. (**A)** The specimen MBM286451 in situ. Laser dots spaced at 33 cm used for measuring dimensions. (**B**) Close-up of branches and polyps in situ. (**C**) The specimen MBM286451 after collection. (**D)** A part of branch with three polyps under a light microscope. (**E**) A single polyp under a light microscope. (**F**) A single polyp under SEM. Scales = 20 cm (**C**), 2 mm (**D**), 1 mm (**E**,**F**).

**Figure 8 biology-10-00588-f008:**
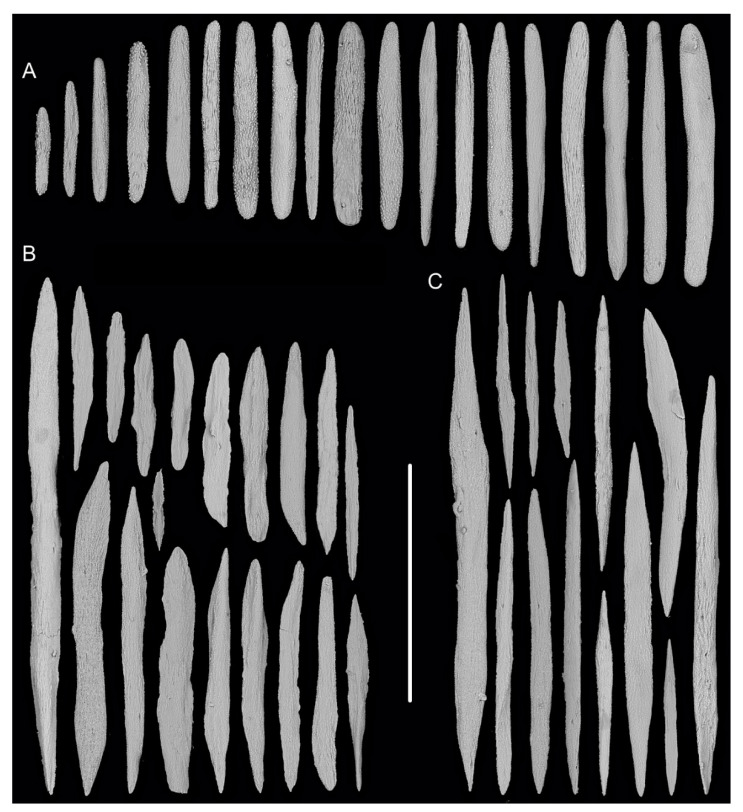
Sclerites of *Iridogorgia magnispiralis* Watling, 2007 (MBM286451). (**A**) Sclerites in tentacle rachis. (**B**) Sclerites in polyp body wall. (**C**) Sclerites in coenenchyme. Scales = 500 μm, all in the same scale.

**Figure 9 biology-10-00588-f009:**
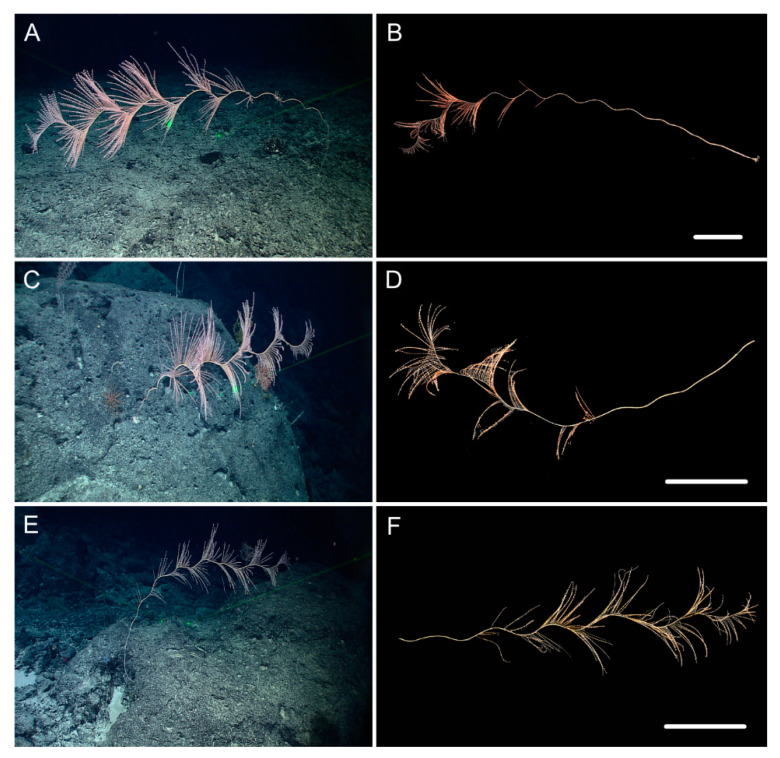
External morphology of *Iridogorgia densispicula* Xu et al., 2020 specimens MBM286446 (**A**,**B**), MBM286447 (**C**,**D**) and MBM286448 (**E**,**F**). (**A**,**C**,**E**) Specimens in situ. (**B**,**D**,**F**) Specimens after collection. Scales = 20 cm (**B**,**D**,**F**).

**Figure 10 biology-10-00588-f010:**
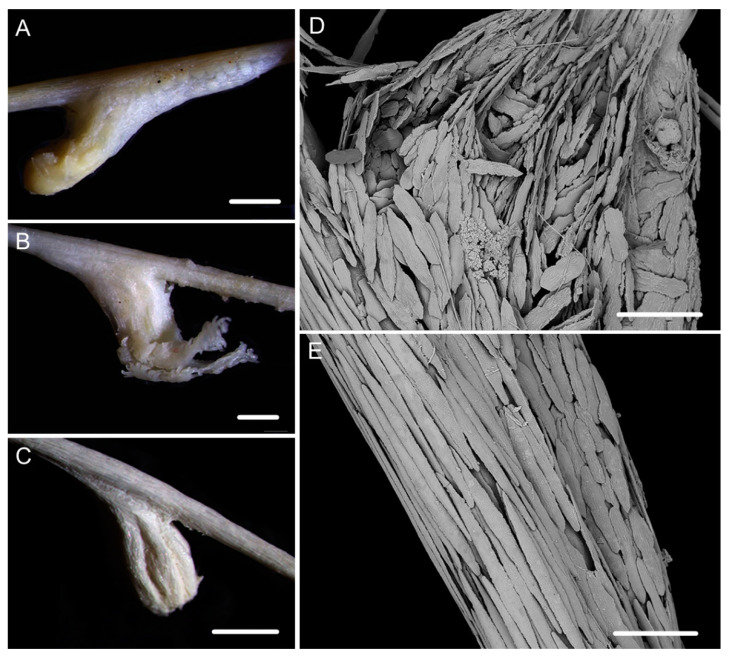
Polyps and sclerites arrangement of *Iridogorgia densispicula* Xu et al., 2020 (**A**–**E**). (**A**–**C**) Single polyps under a light microscope, specimens MBM286446, MBM286447 and MBM286448, respectively. (**D**) Polyp body under SEM. (**E**) Conenechyme and the end of basal body under SEM. Scales = 1 mm (**A**–**C**), 300 μm (**D**,**E**).

**Figure 11 biology-10-00588-f011:**
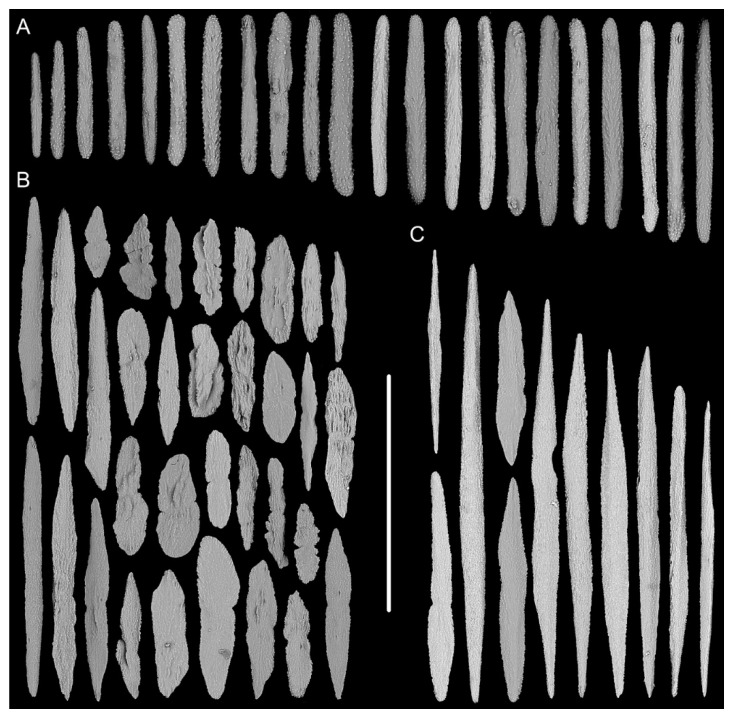
Sclerites of *Iridogorgia densispicula* Xu et al. 2020 specimen MBM286446. (**A**) Sclerites in tentacle rachis. (**B**) Sclerites in polyp body wall. (**C**) Sclerites in coenenchyme. Scales = 500 μm, all in the same scale.

**Figure 12 biology-10-00588-f012:**
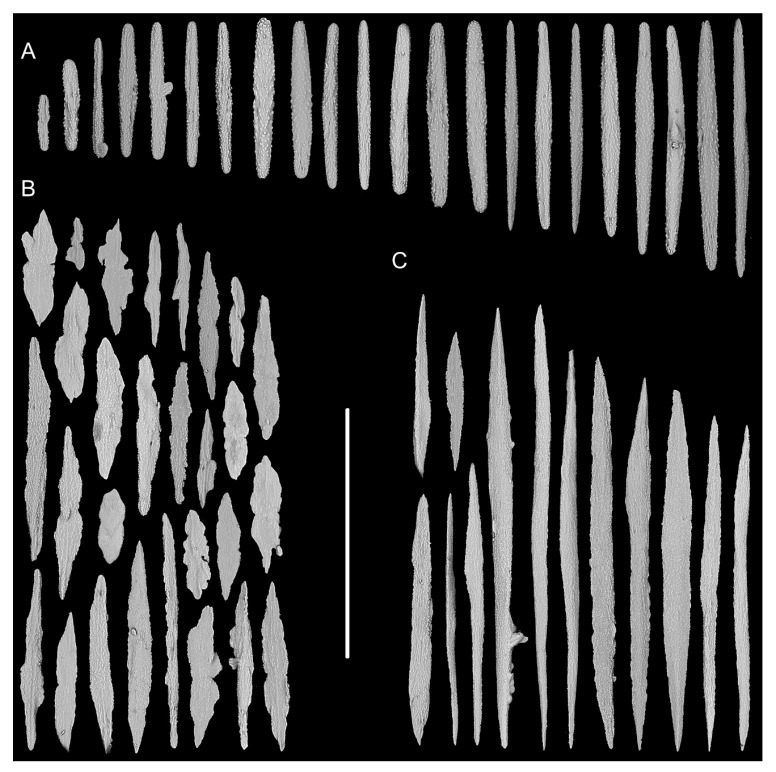
Sclerites of *Iridogorgia densispicula* Xu et al. 2020 specimen MBM286447. (**A**) Sclerites in tentacles. (**B**) Sclerites in polyp body wall. (**C**) Sclerites in coenenchyme. Scales = 500 μm, all in the same scale.

**Figure 13 biology-10-00588-f013:**
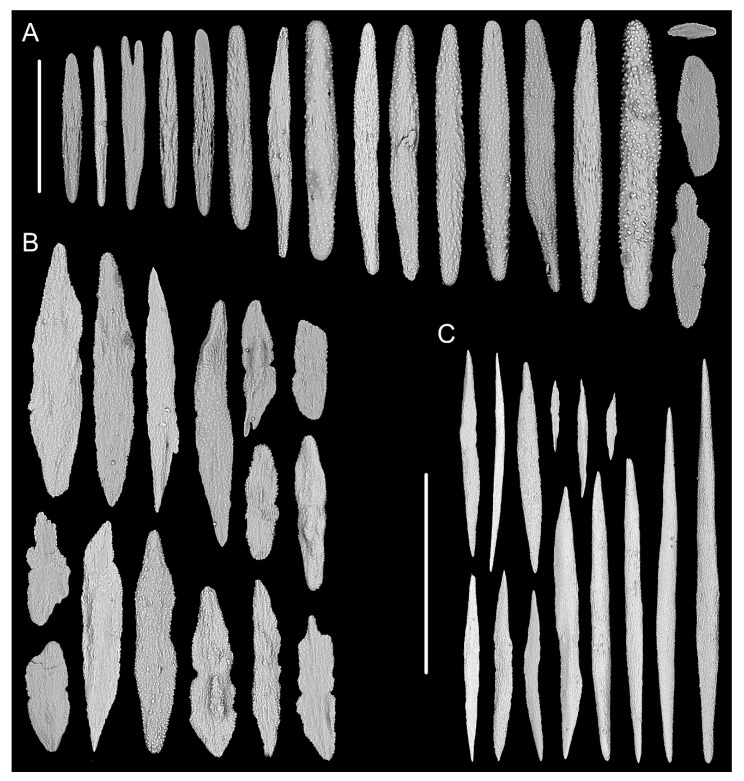
Sclerites of *Iridogorgia densispicula* Xu et al. 2020 specimen MBM286448. (**A**) Sclerites in tentacles. (**B**) Sclerites in polyp body wall. (**C**) Sclerites in coenenchyme. Scales = 200 μm (**A**,**B** in the same scale), 500 μm (**C**).

**Figure 14 biology-10-00588-f014:**
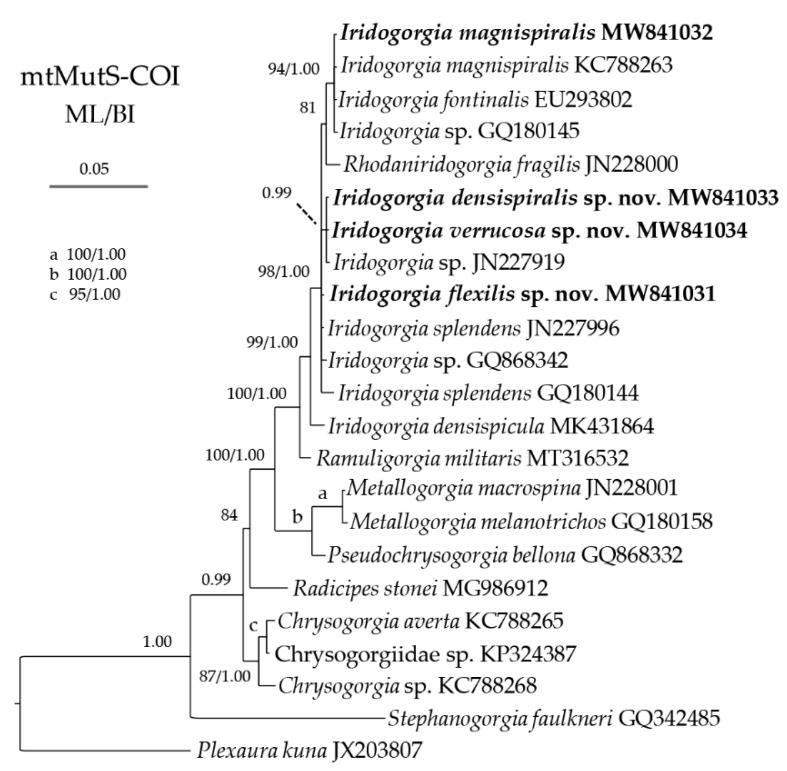
Bayesian inference (BI) tree constructed by the concatenated sequences of mtMutS and COI showing phylogenetic relationships among *Iridogorgia* and *Rhodaniridogorgia* species. The Maximum likelihood (ML) tree is identical to the BI tree in topology. Node support is as follows: ML bootstrap/BI posterior probability. The ML bootstrap <70% or BI posterior probability <0.90 is not shown. Newly sequenced species are in bold. The GenBank accession numbers of the mtMutS sequences were listed next to the species names.

**Figure 15 biology-10-00588-f015:**
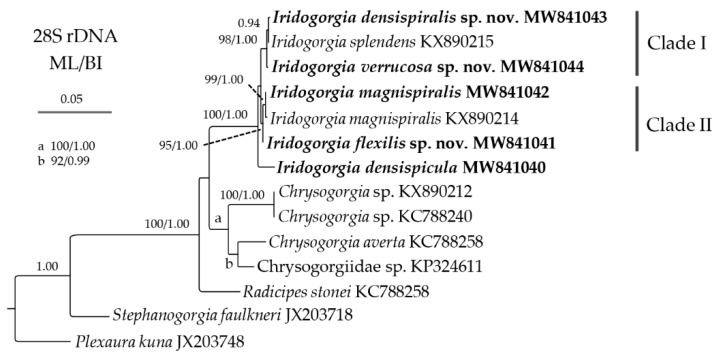
Maximum likelihood (ML) tree constructed by 28S rDNA showing phylogenetic relationships among *Iridogorgia* species. The Bayesian inference (BI) tree is identical with the ML tree in topology. Node support is as follows: ML bootstrap/BI posterior probability. The ML bootstrap <70% or BI posterior probability <0.90 is not shown. Newly sequenced species are in bold. The GenBank accession numbers of the 28S rDNA sequences were listed next to the species names.

**Figure 16 biology-10-00588-f016:**
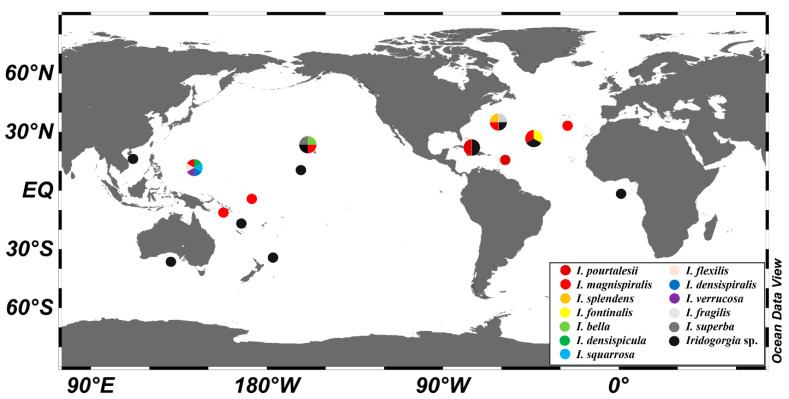
The distributions of the species of *Iridogorgia*, based on the data of references [4,5,6,7,8,10,11,12,13,14,39,40,43].

**Table 2 biology-10-00588-t002:** Comparison of morphological and distributional characters of *Iridogorgia* species. “–” means missing data.

**Characters/Species**	***I. densispicula***	***I. squarrosa***	***I. bella***	***I. fontinalis***	***I. magnispiralis***
Each turn height (cm)	13–18	13–15	–	4	12–20
Helical diameter (cm)	3–4	4–5	–	2.5	1–9
Branch intervals (mm)	3–4	3–4	4	1–2	2–5
Polyp height and width (mm)	2–4/2–4	2–4/1–2	–	1 mm from branch to tentacle base	1–3/2–3
Polyp intervals (mm)	2–7	2–6	7	5–9	2–7
Verrucae	rare	a few	sparse	abundant	abundant
Sclerites in coenenchyme (μm)	spindles usually with two sharp ends: 203–967 × 11–65	spindles usually with two sharp ends: 180–840 × 23–60	–	spindles and a few scales: 397–1024 × 25–74	spindles usually with two sharp ends: 322–1032 × 21–81
Sclerites in polyp bodies (μm)	spindles and scales coarse, usually lobed with irregular shape and rugged and ridged surface in the upper part, and regular and flat in the basal part: 80–627 × 14–110	scales with coarse surface and various shape: 72–569 × 24–150	needle-like or bar-shaped	rods: 175–482 × 27–50	spindles thick and slender, nearly smooth or with many fine warts: 224–383 × 23–57
Sclerites in tentacles (μm)	rods: 137–945 × 11–98	rods: 325–433 × 26–43 scales: 467–805 × 48–66	needle-like or bar-shaped	shorter rods than bodies	rods: 157–548 × 18–78
Distribution	Western Pacific	Western Pacific	near Hawaii	North Atlantic	North Atlantic, near Hawaii, Western Pacific
References	[7], present study	[7]	[6,7,11,13]	[4]	[4,8,11,14], present study
continued					
**Characters/Species**	***I. pourtalesii***	***I. splendens***	***I. flexilis* sp. nov.**	***I. densispiralis* sp. nov.**	***I. verrucosa* sp. nov.**
Each turn height (cm)	–	5	13–15	2.5–3.0	3–4
Helical diameter (cm)	–	1–2	3–4	0.5	1
Branch intervals (mm)	3–6	3–4	3	2–3	1–3
Polyp height and width (mm)	–	less than 1 mm from branch to tentacle base	1–3/2–4	1–2/1–3	2/1–2
Polyp intervals (mm)	5	6–8.5	4–6	3–9	4–6
Verrucae	numerous	numerous	rare	a few	numerous
Sclerites in coenenchyme (μm)	lacking in the inter- polyps	scales and spindles under polyps; absent to rare in the inter-polyps: 274–592 × 24–51	spindles usually with two rounded ends: 322–892 × 25–72	rods and spindles usually with two rounded ends, sometimes sparse in the inter-polyps: 90–523 × 19–65	spindles same as the body wall, usually with two rounded ends: 169–438 × 19–62
Sclerites in polyp bodies (μm)	spindles smooth: average 400	scales with a constriction midway: 143–268 × 29–45	spindles and scales stout and thick with nearly smooth surface: 172–679 × 27–219	spindles, rods and a few elongated scales often thick with sparse and fine warts: 84–347 × 18–60	spindles and elongated scales with sparse and fine warts and irregular edges: 116–450 × 21–65
Sclerites in tentacles (μm)	sparse rods: 200–900	few rods: 169–274 × 27–39	rods: 261–532 × 19–58	rods: 95–442 × 11–52	rods: 160–459 × 17–73
Distribution	North Atlantic	North Atlantic	Western Pacific	Western Pacific	Western Pacific
References	[4,10,11]	[4,6,11]	present study	present study	present study

**Table 3 biology-10-00588-t003:** Genetic distances at 28S rDNA (lower left) and the concatenated mtMutS -28S (upper right) of *Iridogorgia* species.

		1	2	3	4	5	6	7
1	***Iridogorgia densispiralis* sp. nov. MW841043**	-	1.17%	0.31%	3.48%	1.64%	-	-
2	***Iridogorgia flexilis* sp. nov. MW841041**	2.30%	-	1.09%	2.76%	0.46%	-	-
3	***Iridogorgia verrucosa* sp. nov. MW841044**	0.65%	2.13%	-	3.40%	1.56%	-	-
4	***Iridogorgia densispicula* MW841040**	6.08%	4.68%	5.90%	-	3.24%	-	-
5	***Iridogorgia magnispiralis* MW841042**	2.97%	0.65%	2.80%	5.37%	-	-	-
6	*Iridogorgia magnispiralis* KX890214	3.30%	0.98%	3.13%	5.54%	0.32%	-	-
7	*Iridogorgia splendens* KX890215	0.32%	2.30%	0.98%	5.72%	2.97%	3.30%	-

New sequences are in bold.

**Table 4 biology-10-00588-t004:** Interspecific and intraspecific distances at mtMutS of *Iridogorgia* and *Rhodaniridogorgia* species.

		1	2	3	4	5	6	7	8	9	10	11	12	13
1	*Iridogorgia densispiralis* **sp. nov. MW841033**	-												
2	*I. verrucosa* **sp. nov. MW841034**	0.00%	-											
3	*I. densispicula* MK431864	1.11%	1.11%	-										
4	*I. flexilis* **sp. nov. MW841031**	0.15%	0.15%	0.95%	-									
5	*I. splendens* DQ860109, JN227996, JN228005, KC788271, GQ180143	0.16%	0.16%	0.95%	0.00%	0.00%								
6	*I. splendens* GQ180144	0.47%	0.47%	1.33%	0.32%	0.33%	-							
7	*I. squarrosa* MK431865	0.32%	0.32%	1.11%	0.16%	0.16%	0.49%	-						
8	*I. fontinalis* EU293802	0.48%	0.48%	1.27%	0.32%	0.32%	0.66%	0.48%	-					
9	*I. magnispiralis***MW841032** DQ860108, GQ353316, EU268055, JN227997, GQ223116, GQ180141, GQ180142, GQ180140, KC788263	0.47%	0.47%	1.27%	0.32%	0.32%	0.66%	0.47%	0.00%	0.00%				
10	*Iridogorgia* sp. JN227919	0.00%	0.00%	1.11%	0.15%	0.16%	0.47%	0.32%	0.48%	0.47%	-			
11	*Iridogorgia* sp. GQ868342	0.30%	0.30%	1.11%	0.15%	0.16%	0.47%	0.00%	0.48%	0.47%	0.30%	-		
12	*Iridogorgia* sp. DQ297422, GQ180145	0.46%	0.46%	1.27%	0.30%	0.32%	0.63%	0.47%	0.00%	0.00%	0.46%	0.46%	0.00%	
13	*Rhodaniridogorgia fragilis* JN228000	0.70%	0.70%	1.41%	0.47%	0.47%	0.70%	0.93%	0.93%	0.93%	0.70%	0.70%	0.93%	-

New sequences are in bold.

**Table 5 biology-10-00588-t005:** Interspecific and intraspecific distances at COI of *Iridogorgia* and *Rhodaniridogorgia* species.

		1	2	3	4	5	6	7	8	9
1	***Iridogorgia densispiralis* sp. nov. MW841036**	-								
2	***Iridogorgia verrucosa* sp. nov. MW840138**	0	-							
3	***Iridogorgia flexilis* sp. nov. MW841035**	0	0	-						
4	*Iridogorgia splendens* GQ868313, GQ868330, KC788229	0	0	0	0					
5	*Iridogorgia* sp. GQ868323	0	0	0	0	-				
6	***Iridogorgia densispicula* MW841037**	0.20%	0.20%	0.20%	0.20%	0.20%	-			
7	*Iridogorgia fontinalis* GQ868321	0.19%	0.19%	0.19%	0.19%	0.19%	0.39%	-		
8	***Iridogorgia magnispiralis* MW841039**, FJ268639, GQ868318, KC788237	0.19%	0.19%	0.19%	0.19%	0.19%	0.39%	0	0	
9	*Rhodaniridogorgia fragilis* JN227954	0.20%	0.20%	0.20%	0.20%	0.20%	0.39%	0	0	0

New sequences are in bold.

## Data Availability

The specimens described in this study are available at the Marine Biological Museum of Chinese Academy of Sciences (MBMCAS) at Institute of Oceanology, Qingdao, China. Voucher IDs: *Iridogorgia flexilis* sp. nov.: MBM286453; *I. densispiralis* sp. nov.: MBM286454; *I. verrucosa* sp. nov.: MBM286455; *I. magnispiralis* Watling, 2007: MBM286450, MBM286451; *I. densispicula* Xu, Zhan, Li & Xu, 2020: MBM286446, MBM286447, MBM286448. The mtMuts sequences that support the findings of this study have been deposited in NCBI GenBank with the accession codes MW841031 (*I. flexilis* sp. nov.), MW841033 (*I. densispiralis* sp. nov.), MW841034 (*I. verrucosa* sp. nov.) and MW841032 (*I. magnispiralis*). The COI sequences that support the findings of this study have been deposited in NCBI GenBank with the accession codes MW841035 (*I. flexilis* sp. nov.), MW841036 (*I. densispiralis* sp. nov.), MW841038 (*I. verrucosa* sp. nov.), MW841039 (*I. magnispiralis*) and MW841037 (*I. densispicula*). The 28S sequences that support the findings of this study have been deposited in NCBI GenBank with the accession codes MW841041 (*I. flexilis* sp. nov.), MW841043 (*I. densispiralis* sp. nov.), MW841044 (*I. verrucosa* sp. nov.), MW841042 (*I. magnispiralis*) and MW841040 (*I. densispicula*). The new species registration of *Iridogorgia flexilis* sp. nov.*, I. densispiralis* sp. nov. and *I. verrucosa* sp. nov. in Zoobank with LSID: urn:lsid:zoobank.org:act:6BD2C9FC-9CD8-4604-971D-B2EC0BB108B8, urn:lsid:zoobank.org:act:026E2F14-7A66-4558-9D65-B7E3CC0EC8D0 and urn:lsid:zoobank.org:act:E388B7B2-936E-485C-AB38-E7F95DD4F0A4, respectively. The publication LSID: urn:lsid:zoobank.org:pub:273D89B3-0E5C-42C7-9EF7-4526330B92B1.

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
