# Peer review of "Morphological and Molecular Characterization of Five Species Including Three New Species of Golden Gorgonians (Cnidaria: Octocorallia) from Seamounts in the Western Pacific"

_biology, 2021, doi:10.3390/biology10070588_

Round 1
Reviewer 1 Report
In last three years authors have published three paper on chrysogorgiids from the Western Pacific seamounts. I have no doubts that it is crucial to publish new species (and report old species from the area) however, the problem I see - authors publish their data in different journals under practically the same titles, instead of putting all new species of the same genus in the same paper, or publishing them as part 1 and 2 in the same journal. I feel it misleading and hard for final user.
So we have here three new species and new records of two previously described species from unnamed seamounts M5 and M8 at Carolina Ridge ( previously authors described two species from seamount M2 off Mariana trench and seamount M4 from locality they called Carolina plate - but apparently M4 is very close to M8 and based on presented data demonstrate similar species assemblages - at least in terms of Iridogorgia spp)
There are some problems with wording. Simple summary need serious revision both in terms of grammar and wording used. Authors have to have in mind that they report species, and describe new ones. That deep-sea octocorals are not just a component of VME, but represent VME-indicator species. The same problem I see in the introduction. please check carefully lines 51-58. It is very hard to follow logic here and to understand that there are seven described species and three more species are described in the present paper. Need to be seriously re-worded. If authors believe that MOL601-4 belongs to I. squarrosa (I cannot see that any morphological analysis was completed by the authors, it is mentioned as 'may be' squarrosa in the 2020 paper) they have to call it so, if not they have no right to put geographical record. But the species is appearing in the introduction as squarrosa and as Iridogorgia sp./I. squarrosa in the table 1.. and as Iridogorgia GQ868342 in fig.11. It is misleading, authors need to figure out how to present their results to be comprehensive.
It is unclear why authors that promote usage of 28S for phylogeny but did not provide 28S for Iridogorgia squarrosa material described by the same authors in 2020. Apparently it has to be added to the paper. It is very interesting also to add 28S data on Atlantic species (particularly to type species of the genus)
line 145. Authors have to have in mind that genus Iridogorgia was reported in Great Australian Bay, that is technically Indian Ocean (see Supplementary Table in MacIntosh, H., Althaus, F., Williams, A., Tanner, J. E., Alderslade, P., Ahyong, S. T., ... & Wilson, R. S. (2018). Invertebrate diversity in the deep Great Australian Bight (200–5000 m). Marine Biodiversity Records, 11(1), 1-21.
Also see fig 2.8 in Watling et al. 2011 cited by authors [5] for Central East Atlantic (also, please, check, line 596, "Chapter two–biology of deep–water octocorals. " - is absolutely wrong title for the paper)
and Cairns, S. D., Gerhswin, L. A., Brook, F., Pugh, P. R., Dawson, E. W., Ocaña V, O., ... & Fautin, D. G. (2009). Phylum cnidaria; corals, medusae, hydroids, myxozoa. New Zealand Inventory of Biodiversity. Volume 1. Kingdom Animalia: Radiata, Lophotrochozoa, Deuterostomia. - for New Zealand
Iridogorgia magnispiralis was already reported in the Western Pacific (Phoenix area), it is not the first record
Auscavitch, S. R., Deere, M. C., Keller, A. G., Rotjan, R. D., Shank, T. M., & Cordes, E. E. (2020). Oceanographic drivers of deep-sea coral species distribution and community assembly on seamounts, islands, atolls, and reefs within the Phoenix Islands Protected Area. Frontiers in Marine Science, 7, 42.
Table 1. Check museum numbers. There is a mess for YPM and USNM numbers (that were apparently introduced in origonal publications, but authors have to be aware that it is wrong). Number for MNHN material used by authors does not exist anymore - check with MNHN curators. Also please, provide localities and references.
The species that is missed in the Table 1 and has to be included in phylogeny provided by authors is Rhodaniridogorgia fragilis (that as was shown in number of publication cluster inside Iridogorgia group)
Have in mind that descriptions and diagnoses have to follow the same plan and have to be comparable. Comparative diagnosis have to include all hitherto known species of the genus. Please, consider to provide comparative table with all morphological features for all hitherto known species of Iridogorgia, not only known from the area (may facilitate comparison)
Figures, are mostly OK by quality, but in-situ and total views of colonies are too small to be usable. I would prefer to have a photograph of a branch showing density and general arrangement of polyps (on a distance 3-5 cm) for each species. Density of polyps better to include at least to the description.
Fig. 3 has to be split into at least two (two new species need to be presented separately, please provide better resolution figures for 3a-b and 3d-e, so the in situ differences between two new species can be seen, also, provide SEM of individual polyps for densispiralis and verrucosa showing arrangement of sclerites (as done for I.flexilis and I. magnispiralis)
Fig 7 - consider to provide better illustration of in situ and on-deck photographs (bigger and better resolution). Apparently it would be good to make a separate figure for in situ photographs for all species as fig. 1.
it is really misleading that in phylogenetic trees for some species authors use museum numbers and for others accession numbers.
Authors, consider to provide a map with Iridogorgia spp collection sites in the Pacific (including previous records)
line 165 ". Specimen of holotype incomplete" - use instead "holotype is incomplete colony without holdfast"
line 220, 247, 279, 426 please use 'holotype' instead of 'holotype specimen" and 'voucher specimen'
line 301 please provide more accurate identification for stony coral (?scleractinian?)
References. - please check references for redundant capitalization (e.g. [8],[11]), also check that [11] has to be cited as a book
Reviewer 2 Report
Taxonomic papers well written and supported by critical analysis as the present one are original, novel and significant. Although focussed on specialists, they do not reach much attention from other scientists, just in the case they have to cite a species. So, these types of studies are fundamental for biodiversity knowledge, ecological studies and of course management.
The morphologic analysis in this study was very appropriate, I especially like the SEM images and its analysis. The authors present a key for the species in Iridogorgia in the tropical Western Pacific and a complementary molecular analysis that once again demonstrates that without the integration with morphology, boundaries among species are difficult to determine. The description of three new species in the genus is an important contribution to understand its biodiversity and distribution.I recommend the manuscript for publication as it is, but after a minor language review that I am not qualify to do, but I think it is needed and, perhaps native English speaker’s reviewers could provide.
Author Response
Thanks for your comments and encouragement very much.
Reviewer 3 Report
I think this is a very nicely done paper dealing with new species of Iridogorgia from the Western Pacific. The descriptions are well done and the genetic analyses are adequate, but I am not so well-qualified to speak about that.
There are two things I see that might be problematic.
- The key to species includes I. bella, but it is in a couplet after others that deal with the helical coil of the colony. But, at present we do not yet know what the shape of the colony of I. bella actually is. I have been trying to sort that out and have in preparation a redescription of the sclerites and a few other details of I. bella, but unfortunately there is no whole colony or even a significant piece of the colony of I. bella in the museum. So, we are trying to relate the sclerite details of the holotype of I. bella with specimens collected from around Hawaii to see if we can determine the shape of the whole colony. It might be best to leave I. bella out of the key.
- The new species, I. densispiralis looks to me like it may be a species of Rhodaniridogorgia. We have several new species in that genus in our lab in Hawaii. The major difference is supposed to be with the axis, which does not turn in Rhodaniridogorgia, rather the branches spiral around the wavy axis. I do not see Rh. in the molecular genetic analysis, so it would be good to include it. Perhaps the species densispiralis will show that there is no difference between Iridogorgia and Rhodaniridogorgia. If that is the case the diagnosis of Iridogorgia will also need to be changed.
